# Clinical leaders crossing boundaries: A study on the role of clinical leadership in crossing boundaries between specialties

Anoek Braam[1]*, Jeroen D. H. van Wijngaarden[1], Manja Vollmann[2], Carina G. J. M. Hilders[1,3], Martina Buljac-Samardžić[1]

1 Department of Health Services Management & Organisation, Erasmus School of Health Policy and Management, Erasmus University Rotterdam, Rotterdam, Netherlands, 2 Department of Socio-Medical Sciences, Erasmus School of Health Policy and Management, Erasmus University Rotterdam, Rotterdam, Netherlands, 3 Reinier de Graaf Gasthuis, Delft, Netherlands

* braam@eshpm.eur.nl

## Abstract

### Background

Due to the growing number of complex (multimorbid) patients, integrating and coordinating care across medical specialties around patient needs is an urgent theme in current health care. Clinical leadership plays an important role in stimulating coordination both within and between specialty groups, which results in better outcomes in terms of job satisfaction and quality of care.

### Purpose

In this light, this study aims to understand the relation between physicians' clinical leadership and outcomes, focusing on the sequential mediation of relationships and coordination with physicians within their own medical specialty group and from other specialties.

### Methodology

A cross-sectional self-administered survey among physicians in a Dutch hospital (n = 107) was conducted to measure clinical leadership, relational coordination at two levels (medical specialty group and between different specialties), quality of care, and job satisfaction.

### Results

Clinical leadership was related to better quality of care through more relational coordination within the medical specialty group. Clinical leadership was related to more job satisfaction through more relational coordination within the medical specialty group, through more relational coordination between specialties, and sequentially through both kinds of relational coordination.

**Data Availability Statement:** All relevant data are within the paper and its Supporting Information files.

**Funding:** The authors received no specific funding for this work.

**Competing interests:** The authors have declared that no competing interests exist.

## Conclusion

Physicians who act as clinical leaders are important for crossing specialist boundaries and increasing care outcomes.

## Practical implications

To improve multidisciplinary collaboration, managers should encourage clinical leadership and pay attention to the strong relationships between physicians from the same specialty.

## Background

The percentage of people with comorbidities has increased, not only among elderly individuals (above 70) but also in other age categories [1–3]. In the Netherlands, the proportion of adults over 55 who have multiple diseases rose from 22.7% in 2016 to 47% in 2020, 13.6% of adults below the age of 40 suffered from multimorbidity [1]. Complexity of care increased due to the high frequency of multimorbidity, which is often accompanied by problems related to polypharmacy, various treatments, and fragmented medical specialist visits [3–7]. These challenges strongly relate to the fact that the current health care system is still based on a single-disease paradigm that focuses on and subspecializes in single conditions, whereas complex patients have multiple conditions and require an integrated approach involving multiple specialties [6,8,9]. Earlier research stressed that for an integrated approach, structural reorganization is not sufficient [10]. Instead, research suggests that an integrated approach can be supported through relational coordination [11]. According to relational coordination theory, coordination that occurs through frequent, high-quality communication supported by relationships of shared goals, shared knowledge, and mutual respect enables an organization to better achieve desired outcomes [11]. In other words, the effectiveness of coordination is determined by the quality of communication among professionals in a work process, which depends on the quality of their relationships. The quality of their relationships, in turn, reinforces the quality of their communication [11]. However, as a base for the long-term success of integrated care, clinical leadership is also necessary [12]. Clinical leaders are physicians who from an informal position take initiative to, contribute to, and encourage others to improve care. Clinical leaders should serve as role models to demonstrate a clear vision about how to improve patient care and how integrated care can produce these needed improvements [12]. In this study, we aim to explore the associations between clinical leadership, relational coordination, and outcomes in terms of job satisfaction and physicians reported quality of care. Where we anticipate that relational coordination and clinical leadership will both positively influence outcomes, with relational coordination acting as a mediator between clinical leadership and outcomes.

In 2021, Bolton, Logan, and Gittell [13] published a comprehensive review on all studies published from 1991 to 2019 assessing the predictors and outcomes of relational coordination. Their review, based on 233 publications, provides increasing evidence that shared accountability and rewards, shared meetings and huddles, and opportunities to share information and ideas between interdependent physicians can foster teamwork and strengthen relational coordination [13]. A long history of research and guidelines focused on a single disease and hospital structures based on these naturally separated groups of medical specialties [5,6,14,15], provides physicians within the same medical specialty group with the ability to meet the requirements to effectively coordinate care. In the past, accommodating these criteria for

doctors with various medical specialties has received less focus. On the basis of this knowledge, we propose the following hypothesis regarding relational coordination among physicians:

*Hypothesis 1*: *Relational coordination among physicians within their own medical specialty group is stronger than between physicians from different specialties.*

The review by Bolton, Logan, and Gittell [13] also provides evidence that relational coordination among health care professionals is positively associated with quality outcomes (e.g., patient satisfaction, quality of life), efficiency outcomes (e.g., shorter length of stay, reduced costs), and staff outcomes (e.g., job satisfaction, lower burnout rates). Another review by House, Wilmoth, and Kitzmiller [16] showed that relational coordination is positively associated with staff outcomes among healthcare professionals, including higher job satisfaction, better work engagement, lower burnout, lower turnover, and reciprocal learning among health care professionals. Studies that were not covered in these reviews but that have been recently published confirm the positive relationship between relational coordination and employees' well-being (see, for example, Ahmad, Edwin & Bamber [17]; Olaleye [18]). Relational coordination should enable employees to coordinate their work more effectively, which should create the possibility of achieving higher quality of care while also reducing costs [19]. This is why relational coordination appears to be a promising mechanism for raising the standard of care while also addressing financial pressures. In addition, relational coordination can improve job satisfaction by providing professionals with the right resources to accomplish their work. Additionally, it represents high-quality connections, which are associated with job satisfaction [20]. Therefore, we propose the following regarding the relationship between relational coordination among physicians and physician reported quality of care and job satisfaction:

*Hypothesis 2a*: *Physicians reporting higher relational coordination among their own medical specialty group will report higher (a) quality of care and (b) job satisfaction.*

*Hypothesis 2b*: *Physicians reporting higher relational coordination with physicians from other specialties will report higher (a) quality of care and (b) job satisfaction.*

The pressure to integrate and coordinate care across specialties as a result of the rise in complex (multimorbid) patients raises the question of who should take the lead in integrated care in hospitals [6,8]. Some authors suggest that physicians should take the lead in breaking down medical silos [15,21]. Physicians should embrace roles as coordinators, collaborators, and leaders in daily clinical work. Although physicians are used to play such roles within their specific specialist setting, they are now expected to assume responsibilities across disciplines, crossing medical specialist boundaries [22–24]. However, research seems to suggest there are considerable barriers for physicians to take on such roles such as poor interdisciplinary relationships, role conflict, and resistance to change [24]. Clinical leaders, according to Stanley and Stanley [25], are clinicians who are actively involved in clinical care and hold and demonstrate beliefs and values about and passion for high-quality patient care. They are followed because of their visibility in practice and they use their values and beliefs as a driving force to engage in critical problems and face the challenges of clinical care [25]. These clinical leaders are expected to negotiate care plans, balance diverging perspectives in multispecialty teams, and thereby bridge specialist boundaries to provide continuity of care for patients with comorbidities [26–28]. We aim to test this expectation by studying the relationship between clinical leadership and relational coordination among different specialties. Additionally, we expect that these same clinical leadership behaviors will influence coordination and relationships within the medical specialty group. Therefore, the authors hypothesize the following:

*Hypothesis 3*: *Clinical leadership behaviors are positively related to (a) relational coordination among physicians within their medical specialty group and (b) relational coordination among physicians from different medical specialties.*

From the literature on job satisfaction among nurses and physicians, we learn that similar aspects are important for nurses and physicians to be satisfied with their job (salary, autonomy, and interactions with peers) [20,29]. Previous research has shown that nurses who behave as clinical leaders provide higher quality care and are more satisfied with their job [30]. Therefore, it is likely that physicians who show clinical leadership behaviors will also experience these positive effects. Because leadership is deemed necessary to provide effective care coordination, integrate care, and bring about change [28,31,32], we assume that the relationship between clinical leadership and job satisfaction is mediated through relational coordination. Furthermore, studies on how intragroup processes can facilitate more positive intergroup perceptions and experiences show that a strong group relationship and identifying with a group facilitates openness to contact and engagement with others [33]. Based on this knowledge, we propose that effective coordination within one's own medical specialty group is important for crossing boundaries and contributing to the possibility of effective coordination with physicians from different medical specialties. The authors thus hypothesize the following:

*Hypothesis 4: Relational coordination within the medical specialty group and relational coordination among physicians from different specialties sequentially mediate the relationship between clinical leadership and (a) quality of care and (b) job satisfaction (Fig 1).*

## Setting

We conducted our research in a top-clinical hospital. In Dutch health care, there are different kinds of hospitals (general, top-clinical, university) that differ in the care they offer, their expertise, and whether they participate in academic research. A top-clinical hospital is not a university medical center but delivers more complex care and participates more in academic research than a general hospital. Furthermore, the Dutch context involves the existence of the medical specialist company. Many physicians in Dutch hospitals are not salaried workers; they are united with other physicians of the hospital in a medical specialist company. This company has a partnership with the hospital and, together with the board of directors, is responsible for

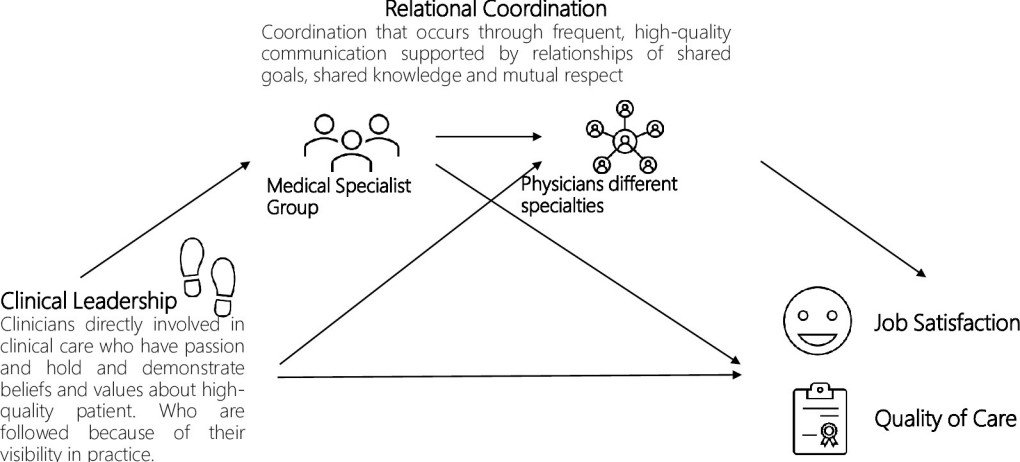

**Fig 1. Representation of the mediation model.**

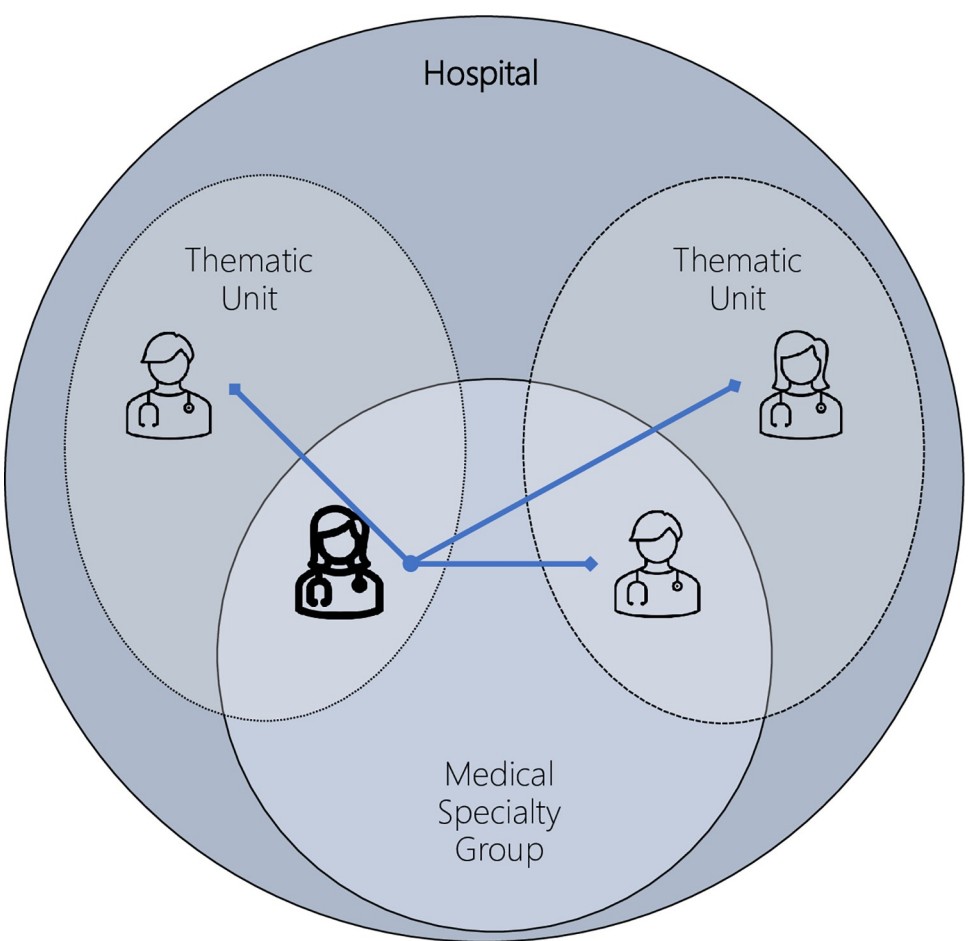

**Fig 2. Representation of the hospital structure for a physician.**

the governance of the hospital, with which they try to reach proper agreements about policy and the care to be provided.

In May 2019, the studied hospital changed its organizational structure. It embedded five accountable multidisciplinary thematic units within its structure: mother and child; chronic care and frail elderly; oncology; acute care; and scheduled care. Within this structure, a single physician belongs to his or her own medical specialty group, belongs to a thematic unit, and is, in general, a physician working in this hospital (Fig 2). Whereas in the past the focus was on medical specialty group silos, emphasis is now placed on the thematic unit. This is reinforced by an organizational communication structure, economic incentives, and dual leadership on the level of the thematic unit. Because of the new structure, physicians from the same medical specialty group may feel stronger connections to different thematic units. For example, some gastroenterologists focus on chronic bowel diseases (e.g., Crohn's disease) and are therefore part of the chronic care unit, while other gastroenterologists focus on gastrointestinal cancer and are part of the oncology unit. Overall, structural change forces and supports thinking in terms of care integration. With this intention, the organizational structure offers opportunities for the integration of care, making it possible in this study to focus on factors important for crossing specialist boundaries without the barrier of an unsupportive organization.

## Method

From October to December 2020, we conducted a cross-sectional survey (S1 Table) among physicians. We approached all physicians, from medical specialists to first-year residents (n = 392). An invitation was sent via email with a direct link to the survey, which was followed by six reminders. Due to a low response rate, we also handed out the survey on paper after the third reminder. In the sixth reminder email, we persuaded doctors (and nurses who received a survey at the same time) to complete the questionnaire with a raffle of 50 champagne bottles among respondents. In total, 139 physicians responded to the survey for a response rate of 35.5%, but 32 of the respondents quit the online survey before answering the first 60% of the questions. Of the 107 physicians (response rate 27.3%), 45.8% identified as female, 44.9% identified as male, 0% as nonbinary, and 9.4% preferred to not reveal their gender or did not answer the question. The majority of the respondents were medical specialists (74.8%) from 27 different specialties (e.g., surgery, radiology, cardiology). The other respondents were junior doctors (4.7%), junior doctors in training (10.3%) or did not reveal their function or answer the question (10.2%). A formal leadership position as manager from an accountable multidisciplinary thematic unit or as coordinator of the medical specialty group in addition to their profession as a physician was held by 23 (21.5%) of the respondents. More than half of the respondents (62.6%) indicated that they had already worked in this hospital for more than six years. We included an opt-out option for the demographic questions to prevent physicians from quitting the survey due to questions about the anonymity of their responses.

## Measurements

**Clinical leadership.**   Physicians' clinical leadership was assessed using a translated version (Dutch) of the Clinical Leadership Survey (CLS) [34]. Patrick and colleagues [34] derived their questionnaire from Kouzes and Posner's (1995) transformational leadership model and adapted the model to reflect general purpose clinical leadership practices and scenarios. The CLS assesses self-perceived transformational leadership behaviors based on 15 items divided into 5 subscales with 3 items each: challenging the process, inspiring a shared vision, modeling the way, enabling others to act, and encouraging the heart. Each item is scored from 1 to 5 (1 = hardly ever to 5 = always). A sample item is "I negotiate with and support members of the interprofessional health care team to help patients achieve their goals". The total clinical leadership score is an average of the 15 items and ranges from 1–5, with higher scores representing more self-reported leadership behavior. In previous research, the CLS has been shown to have a Cronbach's alpha of .86 with Cronbach's alphas for the subscales ranging from .64 to .78 [34]. Our translated Dutch version of the CLS provided an acceptable Cronbach's alpha of .73 for the overall 15-item scale.

**Relational coordination.**   Relational coordination was measured using seven survey questions on a five-point scale (1 = never, 2 = rarely, 3 = occasionally, 4 = mostly, 5 = all the time), including four questions about communication (i.e., frequency, timeliness, accuracy, problem solving) and three questions about relationships (i.e., shared goals, shared knowledge, mutual respect) [19]. These seven questions were asked for two target groups, first for communication and relationships with physicians from the same medical specialty group (e.g., cardiology, surgery) and second for communication and relationships with physicians from different specialty groups (working in our study hospital in the same thematic unit, e.g., frail elderly, oncology). The relational coordination scores were derived by averaging the responses to the items, with higher scores indicating better or more desirable relational coordination [20]. In previous studies, relational coordination has shown a Cronbach's alpha between .80 and .90 [20]. Physicians were asked about communication and relationships with other physicians

from their own medical specialty (α = .87) and physicians from other specialties in the hospital organized within their multidisciplinary thematic unit (α = .88).

**Quality of care.**   To measure quality of care, we used one item that has proven validity from the International Hospital Outcomes Study [35,36]. Physicians were asked to "assess the quality of care from their medical specialty group" on a four-point scale ranging from poor to excellent (1 = poor, 2 = fair, 3 = good, 4 = excellent).

**Job satisfaction.**   A single-item measure of job satisfaction was used. Physicians were asked to rate how satisfied they were with their current job in the hospital on a scale from 0 (completely dissatisfied) to 100 (completely satisfied). The use of this single-item measure is justified by research showing that it is preferred over a sum of items for job satisfaction because multiple items cannot grasp the range of variables that influence job satisfaction, and the single-item measure has shown good reliability and validity [37].

## Analysis

Based on our explanation of structures within the hospital, it could be argued that data were nested within the group structures; however, multilevel analyses were not suitable. A three-level multilevel analysis in which physicians were nested within medical specialty groups and medical specialty groups within thematic units was not suitable because physicians from the same medical specialty were not necessarily nested within the same thematic unit. For two-level multilevel analyses with clustering at the level of the medical specialty group, we conducted the first analysis, the random intercept model, which indicated that there was no clustering effect at the level of medical specialty groups in our data, and continuing multilevel analysis was not appropriate [38]. Furthermore, we had an insufficient number of groups for multilevel analysis; there were only 27 medical specialty groups within the hospital, whereas for multilevel analysis, having 50 or more groups is desirable [39].

Statistical analyses were performed with IBM SPSS Statistics version 27 and PROCESS for SPSS v4.0 [40]. To compare the participants' responses to the relational coordination questionnaire for different collaborations within the medical specialty and between medical specialties, a paired-samples t test was performed. Correlational statistics were used to test the hypotheses on relationships between relational coordination and job satisfaction, relational coordination and quality of care, and clinical leadership and relational coordination. Hypotheses 4a and 4b were tested using Model 6 (sequential mediation model) in PROCESS v4.0 [40]. Two sequential mediation analyses (one for each outcome) were calculated with clinical leadership as the independent variable, relational coordination among physicians from the same medical specialty and relational coordination among physicians from different specialties as sequential mediators, and job satisfaction or quality of care as the dependent variable. The model and path coefficients were estimated using (multiple) regression analyses, while the indirect effects of the independent variable on the dependent variable via the mediator(s) were estimated using bootstrapping with 10,000 bootstrap samples.

## Results

### Relational coordination at different organizational levels

A paired-samples t test was conducted to compare relational coordination scores between physicians from the same medical specialty group (M = 4.42; SD = .52) with relational coordination scores between physicians from different medical specialties (M = 3.87; SD = .53). There was a statistically significant difference between the two scores, $t(97) = 9.60$, $p < .001$ (two-tailed), providing support for Hypothesis 1.

**Table 1. Correlations between study variables.**

| Scale | 2 | 3 | 4 | 5 |
|---|---|---|---|---|
| **1.** Clinical Leadership | .36*** | .46*** | .22* | .33*** |
| 2. Relational Coordination: Physicians from same specialty group | | .38*** | .45*** | .56*** |
| 3. Relational Coordination: Physicians from different specialties | | | .22* | .38*** |
| **4.** Quality of Care | | | | .29*** |
| **5.** Job Satisfaction | | | | |

Significance: * *p* < .05

** *p* < .01

*** p < .001.

Strength: .10 to .29 is weak; .30 to .49 is moderate, .50 to 1.00 is strong.

## Correlations

The relationships between all five variables (clinical leadership, relational coordination medical specialty group level, relational coordination thematic unit level, job satisfaction, quality of care) were investigated using a Pearson product-moment correlation coefficient (see Table 1). All relationships were found to be positive, ranging from weak to strong associations ($0.22 \leq r \geq 0.56$, *p* values < .05). Compared to relational coordination between physicians from different specializations, there are greater correlations between relational coordination between physicians from the same specialist group and job satisfaction and quality of care. However, relational coordination in all its forms shows positive correlations with job satisfaction and quality of care.'

## Sequential mediation

The sequential mediation analyses were based on n = 95 participants with no missing values on the relevant variables (Fig 3). A significant positive total effect of clinical leadership on quality of care was found, indicating that more clinical leadership is associated with a better quality of care when the mediators are not taken into account (ß = .317, t = 2.246, *p* = .027). This effect became nonsignificant when the mediators were included in the model, indicating that clinical leadership is not directly related to quality of care (ß = .098, t = .670, *p* = .505). Rather, a significant positive total indirect effect of clinical leadership on quality of care was found, ß = .16, BC 95% CI [.025,.308]. Further analyses revealed that only one of the three specific indirect effects

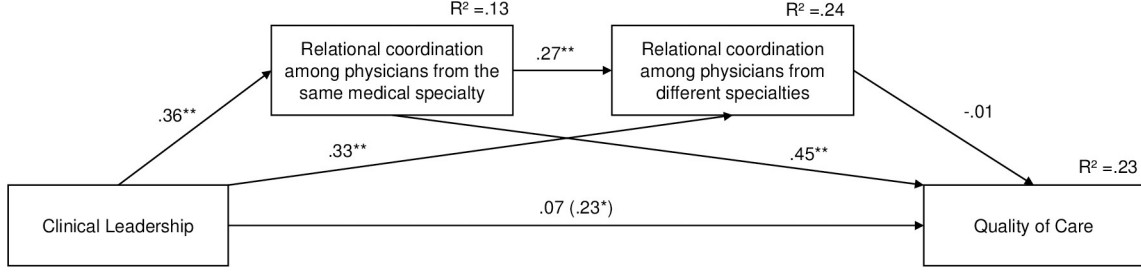

**Fig 3. Results of the mediation model with quality of care as the outcome (H4b).** Standardized path coefficients are reported. The path coefficient in parentheses represents the total effect. Significant indirect effects are indicated by bold printed paths. *p* < .05, **p* < .01, ***p* < .001.

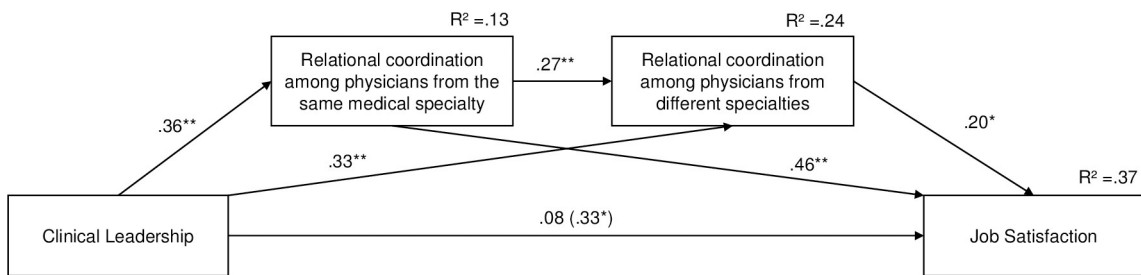

**Fig 4. Results of the mediation model with job satisfaction as the outcome (H4a).** Standardized path coefficients are reported. The path coefficient in parentheses represents the total effect. Significant indirect effects are indicated by bold printed paths. $^*p < .05$, $^{**}p < .01$, $^{***}p < .001$.

of clinical leadership on quality of care was significant. A positive specific indirect effect was found for relational coordination among physicians from the same medical specialty, ß = .16, BC 95% CI [.042,.290], indicating that more clinical leadership is associated with more quality of care through more relational coordination among physicians from the same medical specialty. The specific indirect effects of clinical leadership on quality of care via relational coordination among physicians from different specialties, ß = −.00, BC 95% CI [−.080,.060], and consecutively via both mediators, ß = −.00, BC 95% CI [−.028,.023], were not significant.

The results of the mediation model with job satisfaction as the outcome variable are displayed in Fig 4. A significant positive total effect of clinical leadership on job satisfaction was found (ß = 10.643, t = 3.188, *p* = .001), indicating that more clinical leadership was associated with greater job satisfaction when the mediators were not taken into account. This effect became nonsignificant when the mediators were included in the model (ß = 2.558, t = 3.078, *p* = .408), indicating that clinical leadership is not directly related to job satisfaction. Rather, a significant positive total indirect effect of clinical leadership on job satisfaction was found, ß = .25, BC 95% CI [.080, .398]. Further analyses revealed that all three specific indirect effects of clinical leadership on job satisfaction were significantly positive: first, via relational coordination among physicians from the same medical specialty, ß = .16, BC 95% CI [.002, .284], second, via relational coordination among physicians from different specialties, ß = .07, BC 95% CI [.004, .162], and third, via relational coordination among physicians from the same medical specialty and subsequently relational coordination among physicians from different specialties, ß = .02; BC 95% CI [.001, .076]. These specific indirect effects indicate that more clinical leadership is related to greater job satisfaction through more relational coordination among physicians from the same medical specialty, more relational coordination among physicians from different medical specialties, and consecutively via both.

Most of our hypotheses are supported by the results of the study, except for Hypothesis 4b, which is only partly supported.

## Additional analyses

Independent sample t-tests were performed to compare scores from physicians in a formal leadership position with those who are not, to assess the robustness of the study's findings (S2 Table). The independent t-tests showed no differences between physicians' clinical leadership scores (*t* (98) = 1.35, *p* = .18 two-tailed) for physicians in a formal leadership role (M = 4.05, SD = .38) compared to those not in a formal leadership role (M = 3.92, SD = .50), nor for the other variables used in the analyses.

To assess the robustness of the study's findings, we also conducted separate correlation analyses for the items of relational coordination (i.e., frequent, timely, accurate, problem-

solving, shared goals, shared knowledge, mutual respect) with job satisfaction and quality of care. The results from these analyses (S3 Table) were equivalent to those from the presented main analyses.

## Discussion

The aim of our study was to examine the relationship between physicians' clinical leadership and outcomes (i.e., job satisfaction and quality of care) by focusing on the sequential mediation effect of relational coordination between specialists on two levels: first, relational coordination between physicians from the same medical specialty group (e.g., cardiology, surgery); second, relational coordination between physicians from different specialty groups (working in our study hospital in the same thematic unit, e.g., frail elderly, oncology). Physicians who act as clinical leaders put effort into bridging boundaries by embracing roles as visionary coordinators and collaborators. We expected this to strengthen the relationships and coordination with other physicians, which has been linked in earlier research to improved job satisfaction and quality of care [13]. Our findings show that relational coordination at the group and thematic levels acts as a mediator in the relationship between clinical leadership and job satisfaction. In addition, our findings indicate sequential mediation, in which clinical leadership is first related to relational coordination at the specialty group level, which consecutively impacts relational coordination between different specialties (at the thematic level) and ultimately leads to job satisfaction. Other studies suggest that this sequence may be explained by the fact that more positive intergroup perceptions and experiences lead to more openness to contact with others [33]. This will subsequently be discussed in more detail. For quality of care, only relational coordination at the group level acted as a mediator in the relationship with clinical leadership. The quality measure used represents a physicians' rating of the *"quality of patient care within their own medical specialty group"*. Although multidisciplinary collaboration is deemed necessary for quality of care for a multimorbid patient, physicians might not have considered multidisciplinary care in their answers. Furthermore, our study shows higher levels of relational coordination between physicians within than outside the medical specialty group.

Although the need for collaboration across specialties to meet patients' needs is not being debated, how to achieve this integration in day-to-day practice is [12,41,42]. Earlier research has stressed that structural reorganization to redraw group boundaries is considered insufficient for improved collaboration [10]. Instead, a combination of numerous other strategies may help to improve intergroup relations, such as recognizing and facilitating proactivity, supporting professionals' autonomous motivation, providing formal opportunities for staff collaboration, sending persuasive messages stressing shared values and responsibilities, and differentiating roles [10,43,44]. Our research demonstrates that self-perceived clinical leaders who exhibit behaviors like having deep dialogues with peers are more inclined to collaborate with physicians from other specialties. Clinical leaders appear to help strengthen intergroup relationships.

In addition to the role of clinical leadership in stimulating interdisciplinary cooperation, our research shows the importance of good relations within medical specialty groups. The hospital in which we performed our study aimed to stimulate interdisciplinary cooperation by replacing the existing monodisciplinary units with multidisciplinary units. Initially, it was even suggested that the different specialty groups be dissolved because these groups may hinder a multidisciplinary focus [45]. Traditionally, medical specialty groups play an important role in developing professional identities, producing evidence-based practice, and providing quality control and education [46,47]. As long as specialists derive their identity and security from their medical specialty group, these groups will remain relevant, even in a

multidisciplinary setting. Therefore, it seems that mono- and multidisciplinary physician groups need to coexist and form a network. In the literature, collaboration as networks of interdependent teams that coordinate to achieve shared goals was introduced by Mathieu and colleagues as a multiteam system perspective [48,49]. The work by Amy Edmondson on teaming provides an interesting alternative perspective stating that organizational culture and physicians' mindsets need to be *reframed;* creating awareness among physicians on how their own expertise interacts with other specialties [50]. With the goal of creating fluid, collaborative, interdependent multidisciplinary teams based on patients' needs with a shifting mix of partners across organizational boundaries.

## Limitations

We acknowledge that our research should be interpreted with some caution. First, although the proposed relationships are plausible and theory driven and were consistent with findings from previous studies, the causal direction in the association between the constructs cannot be determined based on cross-sectional data only and requires further study. Second, there is a risk of voluntary response bias because of the low response rate; it is possible that only physicians who felt strongly about the topic decided to participate in our research. Nevertheless, our sample seems to represent the diversity in the physician workforce in a hospital considering the variety represented in medical specialties, physician functions, experience on the job, and experience within the hospital. Third, we used self-reported measures that, despite the guarantee of respondents' anonymity, are subject to various biases, such as social desirability and common method and source bias. However, the risk of common method bias was reduced by using different scales for predictors and outcome variables. Despite these limitations, we believe that our study provides relevant contributions to current scientific and practical debates on clinical leadership, interdisciplinary cooperation, and care coordination. To further understand the collaboration between physicians of various specialties and care coordination, it would be beneficial to conduct similar studies in other (types of) hospitals as the study was only conducted in one. In addition, other outcome metrics, such as patient outcomes, may also contribute to a deeper comprehension. Finally, the current study only used relational coordination between relatively large groups, potentially important insights could be gained by looking at collaboration between physicians per specialty.

## Practical implications

First, our findings suggest that physicians should strive to demonstrate clinical leadership behaviors, as these are associated with increased job satisfaction. In addition, managers should encourage clinical leadership by physicians because the behaviors they exhibit foster relationships among physicians and can strengthen interdisciplinary collaboration.

Second, as seen in this study, there is still potential for a further increase in relational coordination between physicians from different specialties compared to those between physicians from the same specialty. Only the introduction of multidisciplinary structures (as implemented in the study hospital), may not (yet) offer sufficient support to fully commit to multidisciplinary care. As a first step towards future improvement of the quality of collaboration, managers could discuss levels of relational coordination amongst members of the multidisciplinary unit. In addition, focus on multidisciplinary care should be embedded in, amongst others, training and medical quality review, to encourage collaboration and reduce focus on specialist silos.

Third, in contrast to earlier suggestions, our findings show that currently the medical specialty group is like a bird's nest. It provides physicians with a stable base which helps them to

explore and form multidisciplinary collaborations. So, when encouraging multidisciplinary collaboration between physicians' focus should not only be on the multidisciplinary relations, but they should also continue to pay attention to strong connections between physicians from the same specialty.

## Supporting information

**S1 Table. Survey questions and answer options.**
(DOCX)

**S2 Table. Mean scores and differences between physicians in a formal leadership position compared to those not in a formal leadership position.**
(DOCX)

**S3 Table. Correlations matrix for relational coordination items with job satisfaction and quality of care.**
(DOCX)

**S1 Data. Minimal anonymized dataset.**
(SAV)

## Author Contributions

**Conceptualization:** Anoek Braam, Jeroen D. H. van Wijngaarden, Martina Buljac-Samardžić.

**Data curation:** Anoek Braam, Carina G. J. M. Hilders.

**Formal analysis:** Anoek Braam, Manja Vollmann.

**Methodology:** Anoek Braam, Manja Vollmann, Martina Buljac-Samardžić.

**Resources:** Carina G. J. M. Hilders.

**Supervision:** Jeroen D. H. van Wijngaarden, Carina G. J. M. Hilders, Martina Buljac-Samardžić.

**Writing – original draft:** Anoek Braam.

**Writing – review & editing:** Jeroen D. H. van Wijngaarden, Manja Vollmann, Carina G. J. M. Hilders, Martina Buljac-Samardžić.

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
