## [Decision Letter · Decision Letter 0]

13 Jun 2023

PONE-D-23-06594Clinical leaders crossing boundaries: a study on the role of clinical leadership in crossing boundaries between specialties.PLOS ONE

Dear Dr. Braam,

Thank you for submitting your manuscript to PLOS ONE. After careful consideration, we feel that it has merit but does not fully meet PLOS ONE’s publication criteria as it currently stands. Therefore, we invite you to submit a revised version of the manuscript that addresses the points raised during the review process. We are very interested to have reviewed your important study around relational coordination and boundary spanning across clinical leaders.  We believe that improved collaboration and transdisciplinary clinical environments will benefit patients and care givers alike.  However, a main aim of the study around the relational coordination and patient outcomes does not seem to be addressed clearly in the outcomes of the paper.  Furthermore, there does not appear to be direct empirical evidence of meaningful patient outcomes related especially to those with co-morbidities that improve based on the data from the relational coordination scores.  It seems that perhaps the data needs to be analyzed a bit more in depth to answer directly these aims or goals of this study.  It is believed that the data is available in this sample, but not analyzed perhaps in depth enough to answer the main questions directly. Additionally, some participants are leaders and others are not leaders (at least not yet) and therefore this mixed sample could potentially introduce bias into the results that have been provided or that we are now currently asking for more details on.  Please address the potential for bias as all participants are not at leadership positions currently.  Finally, also see the detailed comments from the reviewers of the manuscript.

We look forward to receiving your revised manuscript.

Kind regards,

Philip A. Cola, Ph.D.

Academic Editor

PLOS ONE

Reviewers' comments:

Reviewer's Responses to Questions

**Comments to the Author**

1. Is the manuscript technically sound, and do the data support the conclusions?

Reviewer #1: Partly

Reviewer #2: Yes

2. Has the statistical analysis been performed appropriately and rigorously? 

Reviewer #1: Yes

Reviewer #2: Yes

3. Have the authors made all data underlying the findings in their manuscript fully available?

Reviewer #1: Yes

Reviewer #2: Yes

4. Is the manuscript presented in an intelligible fashion and written in standard English?

Reviewer #1: Yes

Reviewer #2: Yes

5. Review Comments to the Author

Reviewer #1: The study addresses a very important barrier to successful patient care, especially considering patients with multiple comorbidities, and is an important insight into the importance of relational coordination in the hospital setting. In particular, the authors show an important correlation between relational coordination and several important qualities contributing to successful medical care, including care quality, leadership, and job satisfaction both within and between specialties. However, a more in-depth analysis of the survey data, a modification of the study aims/conclusions, and edits to improve clarity would be very useful to improve the manuscript.

Major Points:

The associations between relational coordination and patient outcomes, as stated as the aim of the study, are not addressed. There is no evidence provided of tangible or other reported patient outcomes (particularly, patients with co-morbidities) in different thematic units is correlated to relational coordination score, and the score of physician assessment of their own quality care introduces a major bias despite “proven validity”. Several participants also had formal leadership positions, which may further bias the study results. To prove this claim, the authors should ask question of care satisfaction to their target population – patients with comorbidities – and correlate with the thematic unit responsible for care, if possible.

It would be helpful to have a more in-depth analysis of the correlation of individual questions and points with outcomes, not just as grouped under “relational coordination.” For example, how do the individual components of communication (frequency, timeliness, etc.) correlate with quality of care? Additionally, a differentiation of how responders that hold formal leadership positions responses differ from those that do not could be informative to support several claims in the manuscript.

A more detailed explanation of how to read Table 1 should be provided in the legend to assist with interpretation.

The analysis of the stronger association with the outcome of job satisfaction and relational coordination seems to be biased considering the strong associations between other measured items (i.e. between clinical leadership (group 1) and relational coordination (group 3)) and that these are different based on relational coordination from same specialty group or different specialty groups. More thought should be placed on how those high Pearson product-moment correlations impact the study.

The authors conclude that clinicians should strive to be clinical leaders, which goes hand in hand with job satisfaction. However, there does not seem to be a metric about clinician desire to be an official leader (e.g., have a leadership role in the future, head a department, etc) and how that correlates with relational coordination scores. This would be an important data point to support that claim.

If possible, a supplementary table of questions that were provided to participants should be provided.

Some conclusions made in the manuscript are not entirely supported by the data. For example, the conclusion that this study shows clinical leaders that demonstrate behaviors such as enabling others to act and sharing an inspiring vision will most likely contribute to the improvement strategies is not proven with the data provided. More thought should be given to claims made throughout the text, and they should be modified based off of what the results of the survey support.

Minor Points:

Several hypotheses are provided in the introduction of the manuscript. The study might benefit from consolidating hypotheses into one over-arching hypotheses for clarity and for conciseness.

The main manuscript text can get repetitive, especially in the background section. It could benefit from revision to make points more concise.

The discussion could be expanded to include future directions and additional areas where this survey could be applicable.

Reviewer #2: Review PONE-D-23-06594

Clinical leaders crossing boundaries: a study on the role of clinical leadership in

crossing boundaries between specialties

Thank you for the opportunity to review this important topic.

This article describes the results of a cross-sectional study evaluating the impact of clinical leadership on physicians’ care quality and job satisfaction as mediated by with and between group relational coordination. This study joins a significantly large body of literature on the role of relational coordination in worker wellbeing and work outcomes in general, and an increasingly important area of interest in health care.

Abstract: implications for practice seems a bit odd as the research does not aim to justify continued specialty practice teams. Please consider an implication more closely tied to the importance of promoting relational coordination among physicians and how this might consistently and systematically promoted.

Background:

Line 72. Please add a brief definition of clinical leadership or strong clinical leadership

Line 85-86. Please provide an example or explanation as to why it is more difficult for physicians from different medical specialties to effectively coordinate care

Line 90-91. This information is repeated in the previous paragraph, please revise

Line118 consider choosing just one term: physician versus doctor.

Line 119. Please add a example following this sentence “Although doctors lead from the day they start clinical practice, crossing medical specialist boundaries might be more demanding”

Line 131-133. Suggest moving “a” to precede the first instance of “relational coordination” in the hypothesis so that is reads: Hypothesis 3: Clinical leadership behaviors are positively related to (a) relational coordination among physicians within their medical specialty group and (b) relational coordination among physicians from different medical specialties.

Methods

Line 252. Change ‘respondents’ to participants’

Line 270-271. Change the sentence to read: Participants first agreed to participate in the study before data collection and their identities are kept confidential.

Discussion:

Line 374. please provide elaboration on Amy Edmondson’s work by describing a teaming application appropriate for physician practice and multimorbid care collaboration

Line 387. insert ‘as’ between “such social” to read “ such as social desirability”

Line 390. Suggest adding “ and care coordination” at the end of the sentence

Practical Implications:

As in the abstract, it seems odd to justify the need for specialty group practice. Perhaps adding that more structure/intervention/purposeful socialization is needed between areas of specialty practice in order to take advantage of expertise toward patients’ benefit

6. PLOS authors have the option to publish the peer review history of their article (what does this mean?). If published, this will include your full peer review and any attached files.

Reviewer #1: No

Reviewer #2: **Yes: **Rebecca R. Kitzmiller

---

## [Author Response · Author response to Decision Letter 0]

27 Sep 2023

Please see the attached file: Response to Reviewers

---

## [Editor Report · Decision Letter 1]

30 Oct 2023

Clinical leaders crossing boundaries: a study on the role of clinical leadership in crossing boundaries between specialties.

PONE-D-23-06594R1

Dear Dr. Braam,

We’re pleased to inform you that your manuscript has been judged scientifically suitable for publication and will be formally accepted for publication once it meets all outstanding technical requirements.

Kind regards,

Philip A. Cola, Ph.D.

Academic Editor

PLOS ONE

Additional Editor Comments (optional):

I very much appreciated your detailed and thorough response to the reviewers' comments and my comments as well. I find your responses each to be acceptable. I also appreciated the cleanly edited manuscript with tracked changes and comments. With these changes, I recommend accepting the manuscript as revised.

---

## [Editor Report · Acceptance letter]

2 Nov 2023

PONE-D-23-06594R1 

Clinical leaders crossing boundaries: A study on the role of clinical leadership in crossing boundaries between specialties. 

Dear Dr. Braam:

I'm pleased to inform you that your manuscript has been deemed suitable for publication in PLOS ONE. Congratulations! Your manuscript is now with our production department. 

Kind regards, 

on behalf of

Dr. Philip A. Cola 

Academic Editor

PLOS ONE